# Novel Biocatalysts Based on Bromelain Immobilized on Functionalized Chitosans and Research on Their Structural Features

**DOI:** 10.3390/polym14235110

**Published:** 2022-11-24

**Authors:** Marina G. Holyavka, Svetlana S. Goncharova, Andrey V. Sorokin, Maria S. Lavlinskaya, Yulia A. Redko, Dzhigangir A. Faizullin, Diana R. Baidamshina, Yuriy F. Zuev, Maxim S. Kondratyev, Airat R. Kayumov, Valeriy G. Artyukhov

**Affiliations:** 1Biophysics and Biotechnology Department, Voronezh State University, 1 Universitetskaya Square, 394018 Voronezh, Russia; 2Laboratory of Bioresource Potential of Coastal Area, Institute for Advanced Studies, Sevastopol State University, 33 Studencheskaya Street, 299053 Sevastopol, Russia; 3Metagenomics and Food Biotechnologies Laboratory, Voronezh State University of Engineering Technologies, 19 Revolutsii Avenue, 394036 Voronezh, Russia; 4Kazan Institute of Biochemistry and Biophysics, FRC Kazan Scientific Center of the RAS, 2/31 Lobachevsky Street, 420111 Kazan, Russia; 5Institute of Fundamental Medicine and Biology, Kazan Federal University, 18 Kremlevskaya Street, 420008 Kazan, Russia; 6Laboratory of Structure and Dynamics of Biomolecular Systems, Institute of Cell Biophysics of the RAS, 3 Institutskaya Street, 142290 Pushchino, Russia

**Keywords:** bromelain, enzyme immobilization, carboxymethylchitosan *N*-(2-hydroxypropyl)-3-trimethylammonium chitosan, chitosan sulfate, chitosan acetate

## Abstract

Enzyme immobilization on various carriers represents an effective approach to improve their stability, reusability, and even change their catalytic properties. Here, we show the mechanism of interaction of cysteine protease bromelain with the water-soluble derivatives of chitosan—carboxymethylchitosan, *N*-(2-hydroxypropyl)-3-trimethylammonium chitosan, chitosan sulfate, and chitosan acetate—during immobilization and characterize the structural features and catalytic properties of obtained complexes. Chitosan sulfate and carboxymethylchitosan form the highest number of hydrogen bonds with bromelain in comparison with chitosan acetate and *N*-(2-hydroxypropyl)-3-trimethylammonium chitosan, leading to a higher yield of protein immobilization on chitosan sulfate and carboxymethylchitosan (up to 58 and 65%, respectively). In addition, all derivatives of chitosan studied in this work form hydrogen bonds with His158 located in the active site of bromelain (except *N*-(2-hydroxypropyl)-3-trimethylammonium chitosan), apparently explaining a significant decrease in the activity of biocatalysts. The *N*-(2-hydroxypropyl)-3-trimethylammonium chitosan displays only physical interactions with His158, thus possibly modulating the structure of the bromelain active site and leading to the hyperactivation of the enzyme, up to 208% of the total activity and 158% of the specific activity. The FTIR analysis revealed that interaction between *N*-(2-hydroxypropyl)-3-trimethylammonium chitosan and bromelain did not significantly change the enzyme structure. Perhaps this is due to the slowing down of aggregation and the autolysis processes during the complex formation of bromelain with a carrier, with a minimal modification of enzyme structure and its active site orientation.

## 1. Introduction

Molecules of natural origin are the object of interest for the development of novel drugs and biomaterials due to their physiological activity and the demand for novel therapeutic technologies to be biocompatible and stable in formulations aimed at their preservation and administration [1,2,3,4,5]. Chitosan has been used in various biomedical applications, mainly as a drug carrier for enzymes and peptides, wound-healing accelerator, fat binder, and hemostatic and antimicrobial agent [6]. The molecular structure of this polymer has been offered as an opportunity to add specific mechanical, chemical, or biological characteristics to its conjugations with other molecules [7,8]. The functionalization of chitosan is possible since it possesses functional groups (amino and hydroxy) that are free to bind with other molecules. Chitosan derivatives have demonstrated promising applications in various fields, such as tissue engineering, biomolecule delivery, protection against infections, etc. [9].

Proteases are produced by all organisms—plants, animals, fungi, and bacteria. Bromelain, the protease from the pineapple, is widely used for the treatment of cardiovascular diseases, disorders of blood coagulation and fibrinolysis, infectious diseases, and many types of cancer [10]. Moreover, due to the antiviral, anti-inflammatory, cardioprotective, and anticoagulant activity of bromelain, the enzyme has been suggested as adjunctive therapy for patients with COVID-19 and post-COVID-19. During the spread of new variants of the SARS-CoV-2 virus, such beneficial properties of bromelain could help to prevent the escalation and progression of COVID-19 disease [11]. Bromelain is used in the United States and Europe as an alternative or complementary medication to glucocorticoids, nonsteroidal antirheumatics, and immunomodulatory agents. Very low toxicity ensures its safe use as a remedy for chronic inflammatory diseases, as well as an adjuvant to chemoradiotherapy and perioperative care. Bromelain is capable of enhancing the absorption and tissue permeability of antibiotics after oral, subcutaneous, or intramuscular application. As a result, it can maintain a higher level of a drug in serum and tissue, thus potentiating efficacy and reducing side effects [12].

Stem bromelain (EC.3.4.22.32) is a glycoprotein with one oligosaccharide moiety and one reactive sulfhydryl group per molecule. The highest activity of this enzyme is observed at pH 5.0–8.0 [13]. At a temperature of 21 °C, the aqueous proteolytic activity of bromelain reduces rapidly. Therefore, in a concentrated form (>50 mg/mL), it is stable for a week at room temperature and exhibits minimal inactivation after multiple freeze–thaw cycles [14].

To date, it has been proven that bromelain is well absorbed in the body not only after external but also after oral use and does not have significant side effects even with long-term use. Bromelain has GRAS (Generally Regarded as Safe) status from US federal agencies (CFR 1999, 2009) and can be applied as an enzymatic therapy for humans [15]. Moreover, bromelain has a plant origin, isolated from the extract of *Ananas comosus*, so it is renewable and inexpensive compared with animal-origin enzymes. However, the protease is sensitive to acidity, many chemicals, solvents, and elevated temperatures. Even small conformational changes can reduce enzyme activity, which limits its use in medicine and pharmacology [16,17]. Therefore, one of the most important challenges is the improvement of bromelain stability, mainly by developing new and modifying existing approaches for its stabilization. Among others, the immobilization approach can be used, which was developed to solve the problem of enzyme recovery and reuse [18,19]. Nowadays, a proper immobilization protocol is expected to improve some enzyme features, such as stability, activity, specificity, or selectivity, enlarging the operational conditions [20,21,22]. Moreover, it is possible to couple enzyme immobilization with enzyme purification, saving costs and time with respect to these processes [23].

Enzymes are immobilized using various carriers, including chitosan and its derivatives [24]. For example, bromelain was immobilized on chitosan films from microbial and animal sources and plasticized with glycerol, wherein the highest enzyme immobilization yield (41%) was observed [25]. Chitosan-based nanoparticles, including lactobionic acid-modified chitosan nanoparticles and linoleic acid-modified carboxymethyl chitosan nanoparticles, have also been used for the immobilization of bromelain [26,27,28,29]. Bromelain was covalently immobilized onto the surface of porous chitosan beads without glutaraldehyde [30], with and without alkyl chain spacers of different lengths. The relative activity of immobilized bromelain was found to be high toward a small ester substrate, *N*-benzyl-*L*-arginine ethyl ester (BAEE), but rather low toward casein, a high-molecular-weight substrate [31].

It is known that chitosan is characterized by the limited pH range of its solubility (pKa~6.5) [32], which does not fit with its physiological conditions. It is known that aqueous solutions and solid/liquid interfaces constitute quite different microenvironments that profoundly affect the structure and activity of enzymes. The solid surface exerts an electrostatic field extending several nanometers into the bulk solution. Thus, close to the interface, all electrostatic interactions are modified, altering the behavior of solvent molecules, buffer salts, substrates, products, and the enzyme itself. These effects often lead to enhanced enzyme stability, but also, in most cases, result in some loss of enzyme activity, which has been generally accepted as one of the drawbacks of the immobilization process [33]. The modification of chitosan by tuning its properties and obtaining its water-soluble derivatives can solve this problem. In addition, chitosan has polycationic properties and is characterized by certain antibacterial activities. Its modifications, such as carboxymethylchitosan [34], *N*-(2-hydroxypropyl)-3-trimethylammonium chitosan [35], chitosan sulfate [36], and chitosan acetate [37], also possess antimicrobial activity and retain good biocompatibility, mucoadhesiveness, low toxicity, and other important properties of chitosan. Moreover, chitosan sulfate–lysozyme hybrid hydrogels with fine-tuned degradability have demonstrated sustained, inherent antibiotic and antioxidant activities [38]. In creating new antimicrobial formulations, carboxymethylchitosan, *N*-(2-hydroxypropyl)-3-trimethylammonium chitosan, chitosan sulfate, and chitosan acetate look like promising matrices for bromelain immobilization.

In this work, we show the mechanism of bromelain interaction with the water-soluble derivatives of chitosan—carboxymethylchitosan, *N*-(2-hydroxypropyl)-3-trimethylammonium chitosan, chitosan sulfate, and chitosan acetate—and the proteolytic properties of the obtained compositions.

## 2. Materials and Methods

### 2.1. Materials

Bromelain (B4882) was purchased from Sigma, Burlington, MA, USA, and was used without any treatments. Azocasein (Sigma-Aldrich, Munich, Germany) was used as a substrate in catalytic activity evaluation experiments. Chitosans with molecular weights of 200, 350, and 600 kDa and degrees of deacetylation ranging from 0.73 to 0.85 were obtained from Bioprogress (Shchelkovo, Russia); glycidyltrimethyl ammonium chloride (>90%) and chloroacetic acid (99%) were purchased from Sigma-Aldrich (Munich, Germany). Sodium hydroxide, isopropyl alcohol, methanol, and acetone (all analytical grade); 2% *w/v* aqueous acetic acid solution prepared from glacial acetic acid (analytical grade) and distilled water; 10% *w/v* aqueous sulfuric acid solution fabricated from sulfuric acid (>98%); and distilled water received by Vekton (Saint-Petersburg, Russia) were used in the synthesis of chitosan derivatives.

### 2.2. Synthesis of Chitosan Derivatives

Chitosan derivatives were obtained by known methods with some modifications (see below).

#### 2.2.1. Synthesis of Carboxymethyl Chitosan (ChMC)

Synthesis of carboxymethyl chitosan was carried out according to the following protocol: 3.0 g of chitosan was dispersed in 65 mL of isopropyl alcohol; then, a NaOH aqueous solution was introduced drop by drop for 15 min (NaOH:chitosan repeating link = 13:1 mol). After that, a 15% *w/v* solution of chloroacetic acid in *i*ProOH (CH_2_ClCOOH: chitosan repeating link = 7:1 mol) was added drop by drop to the reaction mixture and stirred for 12 h at room temperature. The resulting precipitate was filtered, suspended, and washed with methanol, and it was dried in a vacuum at 55 ± 2 °C up to constant weight [39]. The yield of products was 79–92%; the degrees of substitution, calculated from the FTIR data, were 0.46, 0.54, and 0.78 for polymers with molecular weights of 600, 350, and 200 kDa, respectively. A significant increase in the degree of substitution for chitosan with a molecular weight of 200 kDa can be explained by the fact that this commercial chemical was in powder form, while the two others were flakes. Carboxymethylation was carried out heterogeneously, so the chitosan form and particle size significantly impacted the resulting products.

#### 2.2.2. Synthesis of *N*-(2-Hydroxy) propyl-3-trimethylammonium Chitosan (HTCCh)

*N*-(2-hydroxy) propyl-3-trimethylammonium chitosan was obtained with the following process: 3.0 g of chitosan was suspended in 30 mL of distilled water for 30 min at 85 ± 2 °C. Then, the calculated amount of glycidyltrimethyl ammonium chloride (GTMAC) (GTMAC:chitosan repeating link = 3:1 mol) was dropwise added to the reaction mixture for 1 h and kept at 85 ± 2 °C for 10 h. The final product was isolated from the reaction mixtures via precipitation in acetone, washed three times with methanol, and dried in a vacuum oven at 55 ± 2 °C to a constant weight [40]. The product yield was 62–74%; the degrees of substitution calculated from FTIR data were 0.24, 0.19, and 0.57 for chitosans with molecular weights of 600, 350, and 200 kDa, respectively.

#### 2.2.3. Synthesis of Chitosan Sulfate, Chitosan Acetate (ChS)

For the preparation of chitosan sulfate, 5.0 g of chitosan was dissolved in 500 mL of 2% *w/v* aqueous acetic acid solution and then 20 mL of 10% *w/v* aqueous sulfuric acid solution and was stirred constantly for 24 h at 25 ± 2 °C. The formed gel was placed in acetone for 5 days, washed three times with methanol, and dried in a vacuum oven at 55 ± 2 °C to a constant weight. The product yield was 85–96%.

#### 2.2.4. Synthesis of Chitosan Acetate (ChA)

For the chitosan acetate synthesis, 5.0 g of chitosan was dissolved in 500 mL of 2% *w/v* aqueous acetic acid solution and stirred constantly for 24 h at 25 ± 2 °C. The final product was isolated from the reaction mixtures via precipitation in acetone, washed three times with methanol, and dried in a vacuum oven at 55 ± 2 °C to a constant weight. The product yield was 67–81%.

### 2.3. Immobilization Procedure

The immobilization of bromelain on the synthesized chitosan derivatives was performed using the complexation approach developed previously [41]. Firstly, 20 mL of enzyme solution (2 mg·mL^−1^ in 0.05 M trisodium borate buffer with a pH of 9.0) was added to 1 g of chitosan derivative (ChMC, HTCCh, ChS, or ChA) and incubated for 2 h at 37 °C. After that, the formed precipitate was washed via dialysis using 50 mM of Tris-HCl buffer with a pH of 7.5 through a cellophane membrane with a 25 kDa pore size until there was an absence of protein in the washing water (controlled spectrophotometrically at λ = 280 nm on an SF-2000 spectrophotometer, LOMO-Microsystems, Saint Petersburg, Russia).

### 2.4. Protein Content Measurement

The protein content in the immobilized enzyme samples was determined using the modified Lowry approach [42]. Before the analysis, the immobilized enzyme was treated with K/Na-tartrate (20 mg·mL^−1^ or 0.7 M) prepared from 1 M NaOH at 50 °C for 10 min to desorb the enzyme from the carrier [43]. The absence of enzyme destruction was controlled by recording its absorption spectrum on a UV-2550PC spectrophotometer (Shimadzu Scientific Instruments Inc., Kyoto, Japan) [44].

### 2.5. Evaluation of Proteolytic Activity of the Immobilized Enzymes

Azocasein was chosen as the substrate for proteolytic activity measurements [45]. Briefly, the sample was dissolved in 200 µL of buffer (50 mM Tris-HCl, pH 7.5), mixed with 800 µL of azocasein solution (0.5% in the same buffer), and incubated for 30 min at 37 °C [46]. Then, 800 µL of 5% trichloroacetic acid solution was added; after 10 min of incubation at 4 °C, the precipitated unhydrolyzed azocasein was removed via centrifugation (3 min 13,000 rpm). The supernatant (1200 µL) was mixed with 240 µL of 1 M NaOH solution, and its optical density was determined at 410 nm. The unit of catalytic activity was considered as the amount of enzyme hydrolyzing 1 µM of azocasein in 1 min (µM·min^−1^·mg^−1^ of protein).

### 2.6. Molecular Docking

The structure preparation of bromelain (PDB ID: 1W0Q, accessed on 31 October 2022 https://www.rcsb.org/structure/1W0Q) for docking and the process of interaction modeling using the Autodock Vina program (Accessed on 31 October 2022 https://sourceforge.net/projects/autodock-vina-1-1-2-64-bit/) were carried out as described in [47]. The structural models of ChMC, HTCCh, ChS, or ChA were drawn in the molecular constructor HyperChem (Accessed on 31 October 2022 https://hyperchem.software.informer.com) and, subsequently, were successively optimized in the AMBER forcefield and quantum-chemically in PM3 (Parametric Method 3). The ligand in the docking procedure had maximal conformational freedom: the rotation of functional groups around all single bonds was allowed. The arrangement of the charges of ChMC, HTCCh, ChS, or ChA and the protonation/deprotonation of their molecules was performed automatically in the MGLTools 1.5.6 package (Accessed on 31 October 2022 https://ccsb.scriptps.edu/mgltools/1-5-6).

### 2.7. Infrared Spectroscopy

Frozen-dried pure protein powders were dissolved in D_2_O. Solid preparations of immobilized proteins on chitosan’s derivatives, ChMC, HTCCh, ChS, and ChA, were washed with buffer solutions in D_2_O. Solutions and solid wet samples were placed on the surface of the ATR working element and equilibrated at 25 °C. The IR spectra of analyzed samples were recorded by an IRAffinity1 spectrometer with an ATR attachment with a single reflection ZnSe working element. The content of protein secondary structures was assessed by fitting absorption spectra in the spectral range 1600–1700 cm^−1^ (amide I absorption band of peptide groups) with the sum of Gaussian components [48]. The position and number of components were determined from the second derivative of the absorption spectra; the fitting was performed using the Fityk 8.0 software.

### 2.8. Bacterial Strains and Biofilm Assays

*Pseudomonas aeruginosa* ATCC 27853 and *Staphylococcus aureus* ATCC 29213 were used for the biofilm assays. Bacteria were grown on the LB medium. To obtain rigid biofilms, bacteria were grown for 48 h under static conditions at 37 °C in 24-well, TC-treated, polystyrol plates (1 mL per well in the basal medium (BM) (glucose 5 g, peptone 7 g, MgSO_4_ × 7H_2_O 2.0 g and CaCl_2_ × 2H_2_O 0.05 g in 1.0 L tap water)) [49]. The mature biofilms were treated for either 3 or 24 h with soluble and immobilized bromelain in PBS, and plates were subjected to crystal violet staining [50].

## 3. Results and Discussions

### 3.1. Bromelain’s Immobilization

As mentioned above, the proper immobilization protocol will improve enzyme characteristics such as stability and half-life, reducing aggregation, and for proteases, it additionally prevents autolysis [51]. However, the enzyme interactions can significantly affect proteolytic activity [52,53].

We estimated the efficiency of complexation for bromelain as a percentage of the adsorbed enzyme and the total and specific activities of the immobilized enzyme compared with the dissolved enzyme (Figure 1). The maximal amount of bromelain was bound with ChS and ChMC, up to 65% and 58% compared with its amount in solution, respectively (Figure 1A).

The bromelain activity assay performed on the azocasein as a substrate showed that the total (U·mL^−1^ of solution) and specific (U·mg^−1^ of protein) proteolytic activity of the enzyme immobilized on ChS, ChMC, and ChA was lower compared with the native enzyme. However, binding with HTCCh led to the bromelain hyperactivation phenomenon (Figure 1B,C). Apparently, the interaction with HTCCh promotes the formation of a more catalytically favorable conformation of bromelain globules modulating the active site and increasing the proteolytic activity.

Depending on the desired use, it is known that carefully selecting chitosan and its derivatives is necessary with attention to the degree of substitution, molecular weight, and purity since these characteristics can significantly affect the mechanical and biological properties of the final product [54,55]. However, in this work, we did not find a direct relationship between the molecular weight of chitosan derivatives and the characteristics of immobilized bromelain we obtained.

Below, we tried to explain the results obtained empirically by studying the mechanism of bromelain complexation with ChMC, HTCCh, ChS, and ChA using in silico (molecular docking) and experimental (FTIR spectroscopy) methods.

### 3.2. Interaction Mechanisms between Bromelain and Chitosan Derivatives

The complexation of enzymes with polysaccharides and their derivatives is a multipronged process that can proceed by using specific bonds and interactions (hydrogen bonds, electrostatic, dipole–dipole, and hydrophobic interactions). The active site orientation regarding the supporting surface plays a critical role, as only properly oriented protease molecules may be accessed by the substrate [56]. Some simple techniques may be used in docking and design studies to account for some of the changes in the conformations of the enzyme during the ligand-binding and complex formation [57].

Like all papain-like proteases, the bromelain molecule is folded into two domains. Domain L is mainly α-helical (α-helices LI, LII, LIII). The key feature of the R domain is its antiparallel β-sheet structure [58]. The R domain also contains two helices: RI and RII, both located on the surface of the protein molecule at opposite ends of the β-sheet structure, which forms the core of this domain [59,60]. The active site of bromelain is located on the border of the L and R domains in a V-shaped cleft formed by Cys26 and His158 [61].

We found that the sorption of bromelain on chitosan derivatives is realized by protein regions located on the border of the L and R domains, including the region of the enzyme’s active site (Table 1, Figure 2 and Figure 3), which results in the modification of their catalytic activity in the immobilized state. In addition, the immobilization involves amino acids included to the α-helices and β-sheets of protein molecules (Table 1), which was confirmed by our FTIR spectroscopy experiments (see below). Moreover, the protein content evaluation (Figure 1A) in the immobilized bromelain formulations is in good agreement with our docking calculations. The protein amount and the protein immobilization yield are higher in bromelain complexes with ChS and ChMC (with which, according to in silico calculations, bromelain forms a greater number of hydrogen bonds—13 and 10, respectively) compared with ChA and HTCCh (with which, according to docking results, bromelain forms a smaller number of hydrogen bonds—8 and 7, respectively). The possible number of hydrogen bonds in the complex is one of the main reasons why the protein immobilization yield is higher. ChMC, ChS, and ChA have hydrogen bonds with His158, which is from the active site of bromelain. This can probably explain the significant decrease in enzyme activity after its immobilization. HTCCh forms only physical interactions with His158, which possibly modulate the structure of the active site of bromelain and lead to the hyperactivation of the enzyme (Figure 1B,C).

To study the hydrogen bonding between bromelain and chitosan derivatives in more detail, we analyzed the FTIR spectra of immobilized samples, including the 3000–3600 cm^−1^ region (Figure 4).

The intensities of the spectra in the 3000–3600 cm^−1^ region of the stretching vibrations of OH groups are normalized to the maximum of the 3300 cm^−1^ band and allow a rough estimate of the intensity distribution of hydrogen bonds: OH groups with stronger hydrogen bonds absorb at lower frequencies. The number of OH groups absorbing below the center frequency of 3300 cm^−1^ is the smallest in the bromelain/HTCCh system, followed by ChMC, ChA, and ChS in ascending order. The number of OH groups with weak or single hydrogen bonds absorbing above 3300 cm^−1^ is the highest in the bromelain/ChS system, followed in descending order by ChA, HTCCh, and ChMC. It should be noted that it is not possible to exactly estimate the number of hydrogen bonds between the protein and chitosan matrices: the number of these bonds can only be estimated in tens of pieces, while the total number of OH bonds in both the protein and chitosan can reach many hundreds—these differences are completely lost in measurement errors. The above-cited experimental ratios reflect the rearrangement of the entire system of hydrogen bonds as a result of the formation of protein–chitosan complexes, including the secondary structure of the protein and the conformation of the polymer [62], as well as the molecules of hydration water included in their structure.

Figure 5 and Table 2 show the results of an FTIR estimation of the secondary structure of bromelain immobilized on chitosan derivatives: ChMC, ChS, and ChA. It can be seen that after complexation, the number of α-helix contents remains at the same level, the portion of β-sheets significantly increases, and the number of other structures notably decreases.

For the complex with HTCCh, such calculations could not be carried out because bromelain was weakly retained on the HTCCh matrix and was washed off with buffer. It was found that complexation with HTCCh did not significantly change the enzyme structure since its structure in the supernatant after washing with buffer was close to the structure of bromelain in solution (without any complexation). Despite the low stability of the bromelain complex with HTCCh, the hyperactivation of the enzyme can be observed. Perhaps this is due to the slowing of aggregation and autolysis processes during the complex formation of bromelain with HTCCh, with minimal modification of enzyme structure and its active site orientation.

Finally, we compare the activity recovery of immobilized bromelain obtained in this work with our previously published results (Figure 6). Earlier, we studied bromelain immobilization on chitosan (Ch) [46], 2-(4-acetamido-2-sulfanilamide) chitosan (SACh) [41], and chitosan copolymer with poly-*N*-vinylimidazole (Ch-g-PVI) [63]. As can be seen, the best activity recovery is achieved for bromelain immobilized on HTCCh and SACh, while for other proposed matrices, its value is dramatically low. Only under the immobilization of chitosan was the phenomenon of bromelain hyperactivation observed. Thus, HTCCh-immobilized bromelain is a promising biocatalayzer for biomedical and biotechnological applications.

### 3.3. Antibiofilm Properties of Immobilized Bromelain

Various proteases have been suggested as promising antibiofilm tools [64,65], including the cysteine plant proteases ficin and papain in soluble and chitosan-immobilized forms [66,67,68,69,70,71]. Therefore, the ability of both soluble and immobilized bromelain to disrupt the bacterial biofilms was assessed. For that, 24 h old biofilms were obtained on the surface of 24-well plates washed and treated for either 3 or 24 h with soluble bromelain (0.1 and 0.5 mg·mL^−1^) or enzyme immobilized on water-soluble derivatives of chitosan—carboxymethylchitosan, *N*-(2-hydroxypropyl)-3-trimethylammonium chitosan, chitosan sulfate, and chitosan acetate. The amount of heterogeneous enzyme corresponded to 0.1 and 0.5 mg·mL^−1^ of soluble enzyme, and as a negative control, pure carriers were taken in similar concentrations. The residual biofilms were subjected to crystal violet staining.

Soluble bromelain destroyed biofilms formed by both *P. aeruginosa* and *S. aureus* (See Figure 7). Among all the variant derivatives of chitosan, only carboxymethylchitosan demonstrated relevant results, and carboxymethylchitosan-immobilized bromelain led to a significant reduction in the biofilms of both pathogens. *N*-(2-hydroxypropyl)-3-trimethylammonium chitosan, chitosan sulfate, and chitosan acetate by itself and with immobilized protease led to false-negative results, apparently because of the unspecific adherence of plate surfaces (Figure 8 and Figure 9). Taking into account the significant relevance of both *P. aeruginosa* and *S. aureus* on topical surgery wounds and burns, the antibiofilm activity of immobilized bromelain could be a promising approach to coat wound-dressing materials to disrupt and prevent the biofouling of wound-dressings and wounds.

## 4. Conclusions

Taken together, our results show that ChS and ChMC form the highest number of hydrogen bonds with bromelain (thirteen and ten, respectively) compared with eight and seven H-bonds for ChA and HTCCh. Apparently, it is one of the main reasons why the protein immobilization yield is higher at complexation with ChS and ChMC (up to 58% and 65%). Moreover, ChMC, ChS, and ChA have H-bonds with His158, which is from the active site of bromelain. This can explain the significant decrease in enzyme activity after its immobilization. HTCCh displays only physical interactions with His158, which possibly modulate the structure of bromelain’s active site and lead to enzyme hyperactivation of up to 208% of the total activity (U·mL^−1^ of solution) and up to 158% of the specific activity (U·mg^−1^ of protein). From the FTIR study, it was also found that complexation with HTCCh does not significantly change the bromelain structure. The carboxymethylchitosan-immobilized bromelain provides a significant reduction of the biofilms of both *P. aeruginosa* and *S. aureus*, bacteria often present on topical surgery wounds and burns, suggesting that immobilized bromelain is a promising tool for disrupting and preventing the biofouling of wound-dressing materials and wounds.

## Figures and Tables

**Figure 1 polymers-14-05110-f001:**
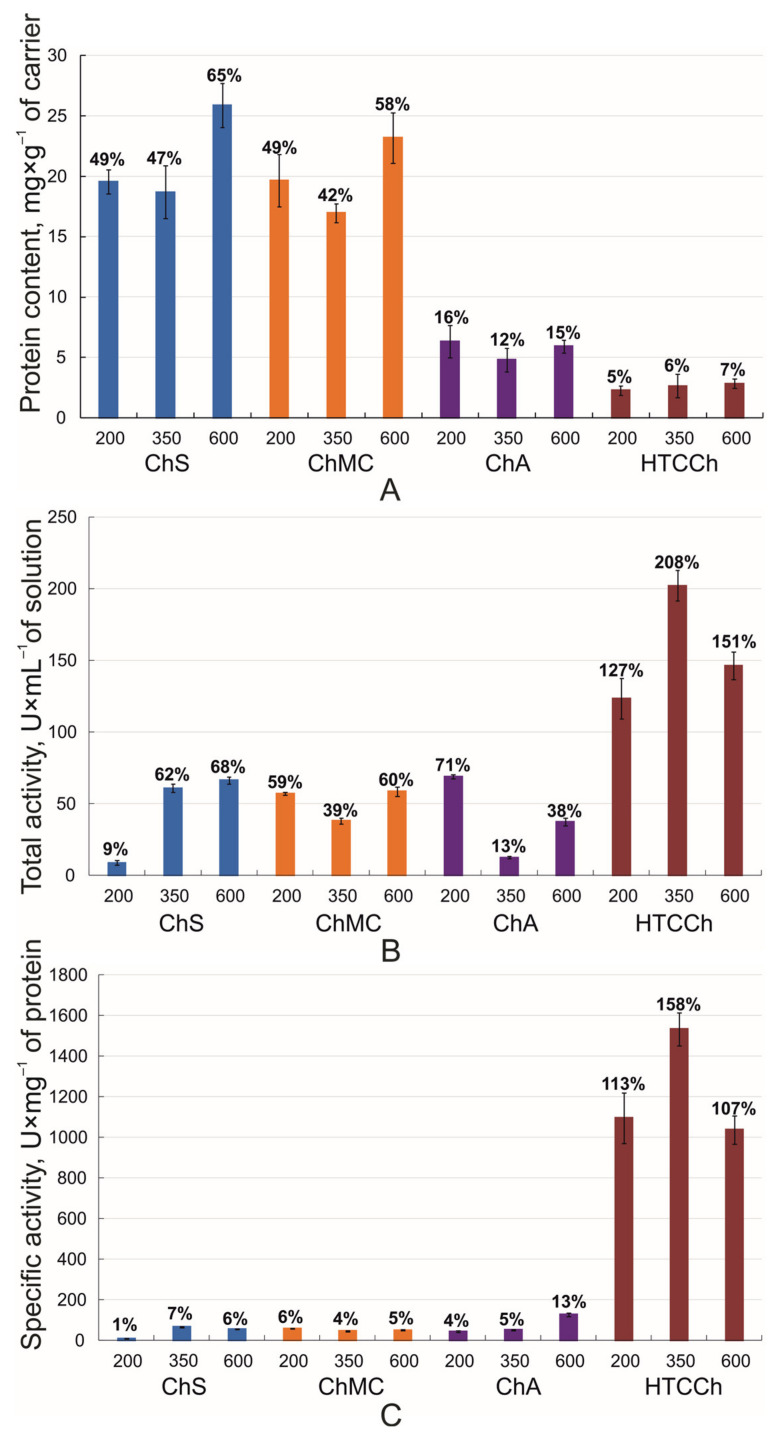
Characteristics of immobilized bromelain: (**A**) Protein content (mg·g^−1^ of carrier) in enzyme complexes with ChS, ChMC, ChA, and HTCCh; (**B**) total activity (U·mL^−1^ of solution) of enzyme complexes with ChS, ChMC, ChA, and HTCCh; (**C**) specific activity (U·mg^−1^ of protein) of enzyme complexes with ChS, ChMC, ChA, and HTCCh. The efficiency of complexation for bromelain expressed as a percentage of the adsorbed enzyme compared with its amount in solution (**A**), the total activity of immobilized enzyme compared with soluble enzyme (**B**), and the specific activity of the immobilized enzyme compared with the dissolved enzyme (**C**) are indicated above the bars. All experiments were performed eight times, and the results represent mean ± confidence interval.

**Figure 2 polymers-14-05110-f002:**
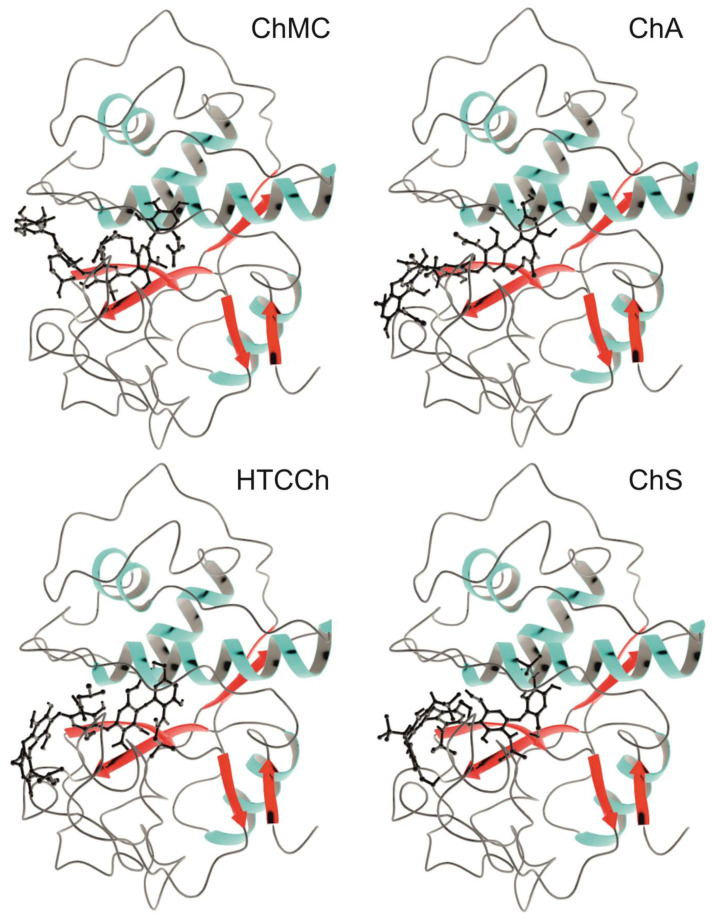
Molecular docking results. Topology of bromelain complexes with ChMC, HTCCh, ChS, and ChA.

**Figure 3 polymers-14-05110-f003:**
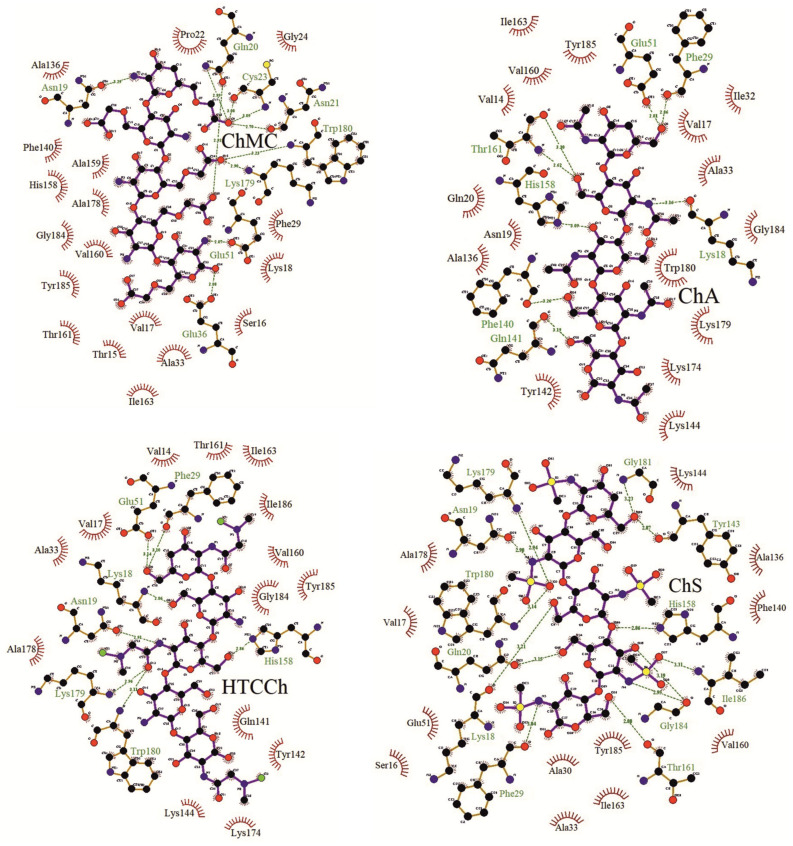
Molecular docking results. Bonds (indicated by dashed lines) and interactions between molecules of bromelain and chitosan derivatives: ChMC, HTCCh, ChS, and ChA.

**Figure 4 polymers-14-05110-f004:**
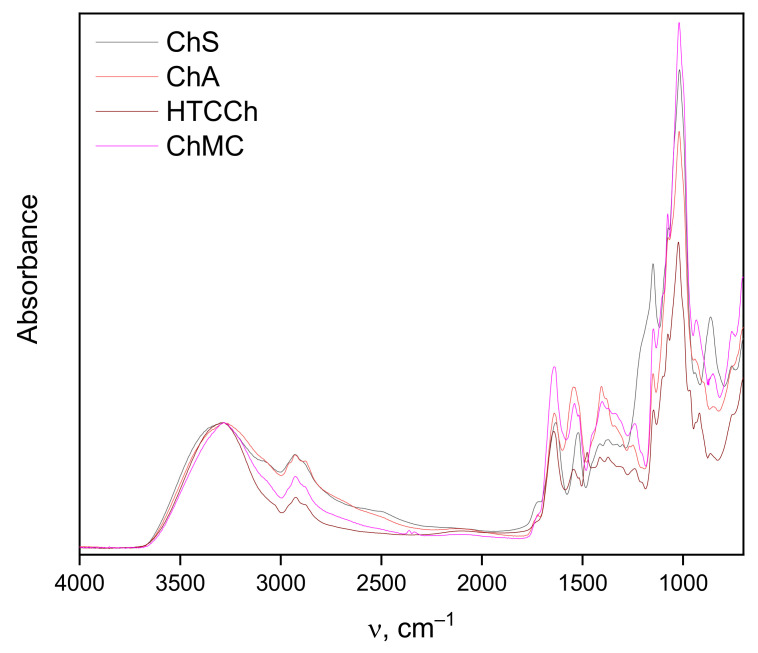
FTIR spectra of dry preparations of bromelain on ChS, ChA, HTCCh, and ChMC matrices.

**Figure 5 polymers-14-05110-f005:**
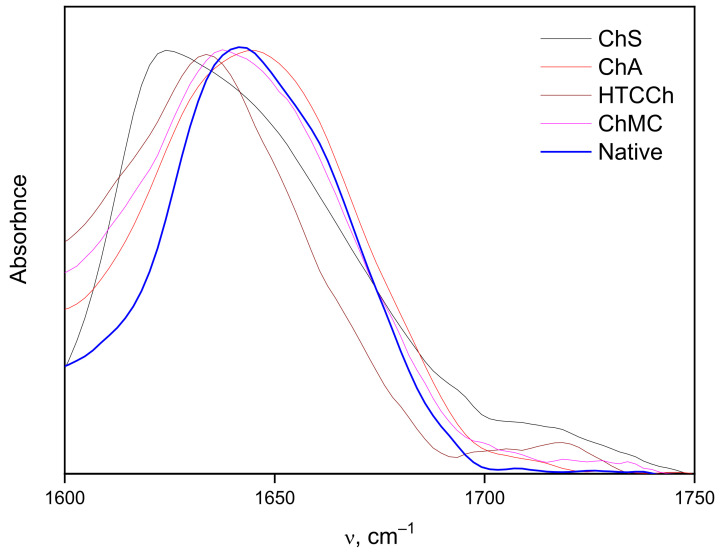
Amide I spectra of bromelain and chitosan derivative complexes are normalized to the maximum.

**Figure 6 polymers-14-05110-f006:**
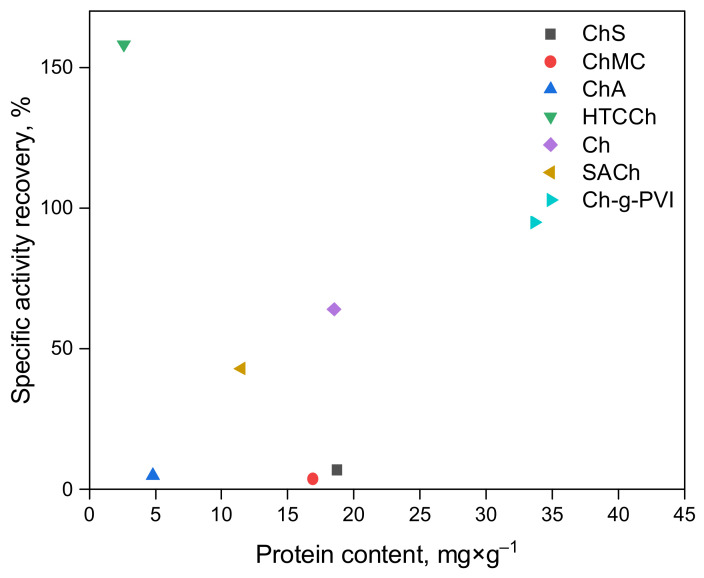
Comparison of specific activity recovery (%) of bromelain immobilized on different chitosan derivatives (values are given for derivatives with a molecular weight of 350 kDa).

**Figure 7 polymers-14-05110-f007:**
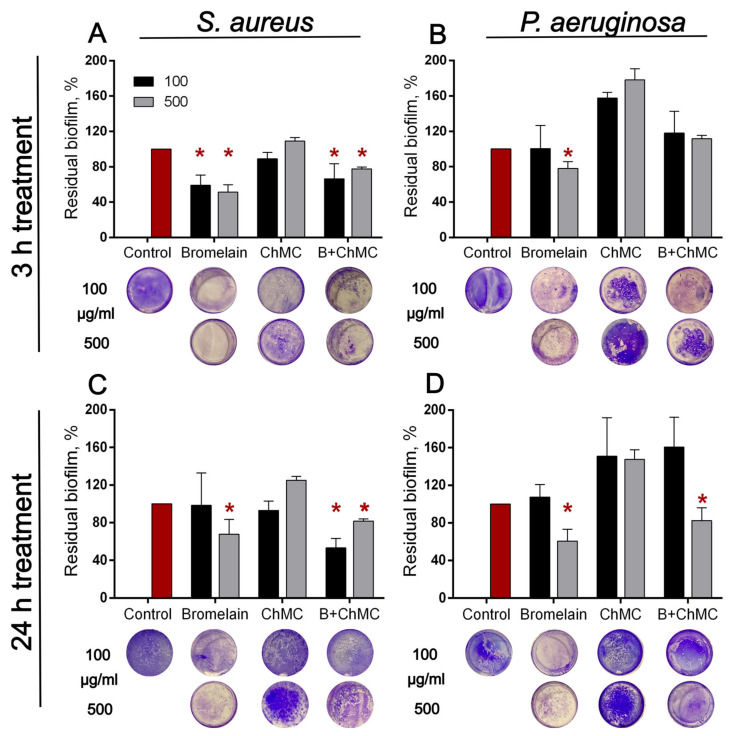
Effect of dissolved bromelain and enzyme immobilized on carboxymethylchitosan (ChMC) on (**A**,**C**) *S. aureus* and (**B**,**D**) *P. aeruginosa* biofilms. The 48 h old biofilms were gently washed and treated for 3 (**A**,**B**) or 24 h (**C**,**D**) with enzymes at concentrations of 0.1 and 0.5 mg·mL^−1^. The amount of heterogeneous enzyme corresponded to 0.1 and 0.5 mg·mL^−1^ of the soluble enzyme, and as a negative control, pure carriers were taken in similar concentrations. Residual biofilms were quantified with crystal violet staining. Asterisks (*) denote statistically significant differences between untreated and treated samples (*p* < 0.05). Untreated wells were considered to be 100%.

**Figure 8 polymers-14-05110-f008:**
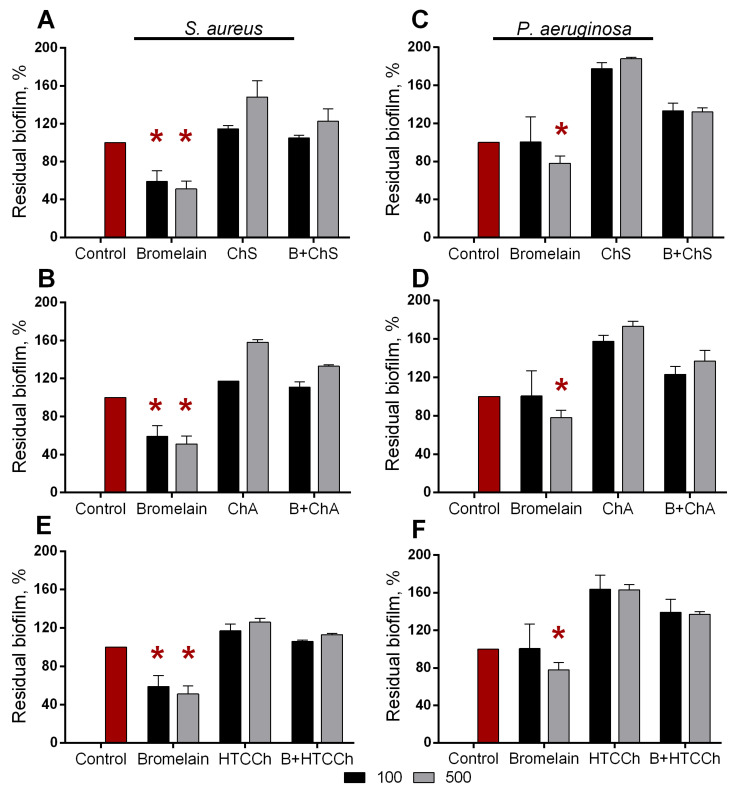
Effect of soluble bromelain and enzyme immobilized on *N*-(2-hydroxypropyl)-3-trimethylammonium chitosan (HTCCh), chitosan sulfate (ChS), and chitosan acetate (ChA) on (**A**,**B**,**E**) *S. aureus* and (**C**,**D**,**F**) *P. aeruginosa* biofilms. The 48 h old biofilms were gently washed and treated for 3 h with enzymes at concentrations of 0.1 and 0.5 mg·mL^−1^. The amount of heterogeneous enzyme corresponded to 0.1 and 0.5 mg·mL^−1^ of the soluble enzyme; pure carriers in similar concentrations were taken as a negative control. Residual biofilms were quantified with crystal violet staining. Asterisks (*) denote statistically significant differences between untreated and treated samples (*p* < 0.05). Untreated wells were considered to be 100%.

**Figure 9 polymers-14-05110-f009:**
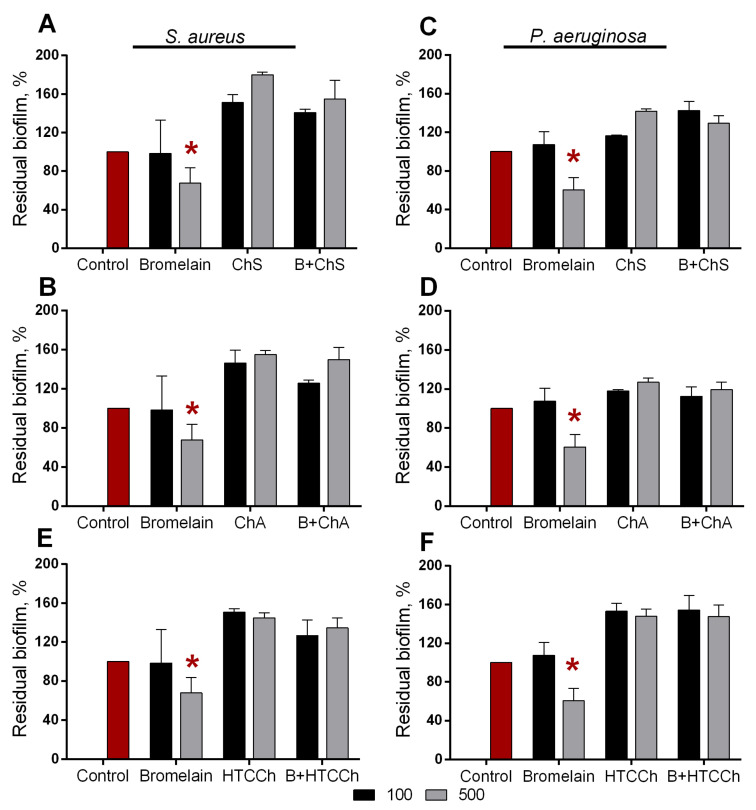
Effect of soluble bromelain and enzyme immobilized on *N*-(2-hydroxypropyl)-3-trimethylammonium chitosan (HTCCh), chitosan sulfate (ChS), and chitosan acetate (ChA) on on (**A**,**B**,**E**) *S. aureus* and (**C**,**D**,**F**) *P. aeruginosa* biofilms. The 48 h old biofilms were gently washed and treated for 24 h with enzymes at concentrations of 0.1 and 0.5 mg·mL^−1^. The amount of heterogeneous enzyme corresponded to 0.1 and 0.5 mg·mL^−1^ of the soluble enzyme; pure carriers in similar concentrations were taken as a negative control. The residual biofilms were quantified with crystal violet staining. Asterisks (*) denote statistically significant differences between untreated and treated samples (*p* < 0.05). Untreated wells were considered to be 100%.

**Table 1 polymers-14-05110-t001:** Bromelain amino acids interacting with ligands *.

Affinity, kcal/mol	Amino Acids Forming	
	Hydrogen Bonds and Their Lengths, Å	Any Other Interactions
Amino acids of bromelain that form bonds and interactions with ChMC		
−7.8	Asn19, 3.29 Å; Gln20, 2.92 and 2.99 Å; Asn21, 2.70 Å; Cys23, 3.00 and 3.09 Å; Glu36 (αLI), 2.88 Å; Glu51 (αLII), 2.87 Å; Lys179, 2.90 Å; Trp180, 3.23 Å	Thr15, Ser16, Val17, Lys18, Pro22, Gly24, Phe29 (αLI), Ala33 (αLI), Ala136, Phe140, **His158** (βR), Ala159 (βR), Val160 (βR), Thr161 (βR), Ile163 (βR), Ala178, Gly184, Tyr185
Amino acids of bromelain that form bonds and interactions with HTCCh		
−8	Lys18, 2.86 Å; Asn19, 2.91 Å; Phe29 (αLI), 3.10 Å; Glu51 (αLII), 3.14 Å; **His158** (βR), 2.86 Å; Lys179, 2.96 Å; Trp180, 3.11 Å	Val14, Val17, Ala33 (αLI), Gln141, Tyr142, Lys144, Val160 (βR), Thr161 (βR), Ile163 (βR), Lys174, Ala178, Gly184, Tyr185, Ile186
Amino acids of bromelain that form bonds and interactions with ChS		
−9.0	Lys18, 3.21 Å; Asn19, 2.90 Å; Gln20, 3.15 Å; Phe29 (αLI), 3.04 Å; Tyr143, 2.87 Å; **His158** (βR), 2.86 Å; Trp161 (βR), 2.80 Å; Lys179, 2.94 Å; Trp180, 3.14 Å; Gly181, 3.23 Å; Gly184, 2.95 and 3.19 Å; Ile186, 3.31 Å	Ser16, Val17, Ala30 (αLI), Ala33 (αLI), Glu51 (αLII), Ala136, Phe140, Lys144, Val160 (βR), Ile163 (βR), Ala178, Tyr185
Amino acids of bromelain that form bonds and interactions with ChA		
−9.4	Lys18, 3.16 Å; Phe29 (αLI), 2.86 Å; Glu51 (αLII), 2.81 Å; Phe140, 3.26 Å; Gln141, 3.19 Å; **His158** (βR), 3.09 Å; Thr161(βR), 2.62 and 3.30 Å	Val14, Val17, Asn19, Gln20, Ile32 (αLI), Ala33 (αLI), Ala136, Tyr142, Lys144, Val160 (βR), Ile163 (βR), Lys174, Lys179, Trp180, Gly184, Tyr185

* Active site residues are bold; protein secondary structure elements are in brackets.

**Table 2 polymers-14-05110-t002:** Secondary structure (in%) of bromelain immobilized on chitosan derivatives: ChMC, ChS, and ChA.

Structure	Sample			
	Bromelain in Solution	Bromelain with ChMC	Bromelain with ChS	Bromelain with ChA
α-helix	15	13	13	19
β-sheet	29	43	52	40
other	56	44	35	41

## Data Availability

The data presented in this study are available on request from the corresponding author.

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
