# Peer review of "Novel Biocatalysts Based on Bromelain Immobilized on Functionalized Chitosans and Research on Their Structural Features"

_polymers, 2022, doi:10.3390/polym14235110_

Round 1

Reviewer 1 Report

The presented results are interesting and have potential to be useful in development and various applications of novel biocatalysts in future. 

In introduction it would be beneficial to provide more citations related to applications of bromelain jointly with chitosan derivatives and to justify the selection of chitosan derivatives which have been selected for synthesis in the presented manuscript.

Hydrogen bonds are mentioned in the text, but not discussed in detail and full FTIR spectra including the region 3000-3600 cm-1 are not provided and band 3000-3600 cm-1 was not analyzed at all.

Chitosans with 3 different molecular weights were studied and presented but the effect of molecular weight of chitosan on the results has not been clearly discussed.

It would be beneficial to discuss future perspectives in Conclusions in more detail

Author Response

We would like to thank the Reviewers for taking their time to read our manuscript. In the view of the constructive criticism by the Reviewers, we have revised the manuscript considerably. All our corrections are highlighted in yellow in the text of the article

In the following, we response to particular concerns raised by the Reviewers in a step-by-step manner.

Reviewer 1, comment 1

The presented results are interesting and have potential to be useful in development and various applications of novel biocatalysts in future. 

In introduction it would be beneficial to provide more citations related to applications of bromelain jointly with chitosan derivatives and to justify the selection of chitosan derivatives which have been selected for synthesis in the presented manuscript.

Response

Citations related to applications of bromelain jointly with chitosan derivatives were added to the introduction section.

Reviewer 1, comment 2

Hydrogen bonds are mentioned in the text, but not discussed in detail and full FTIR spectra including the region 3000-3600 cm-1 are not provided and band 3000-3600 cm-1 was not analyzed at all.

Response

Relevant data has been added to the text of the article.

Reviewer 1, comment 3

Chitosans with 3 different molecular weights were studied and presented but the effect of molecular weight of chitosan on the results has not been clearly discussed.

It would be beneficial to discuss future perspectives in Conclusions in more detail

Response

In this work, we did not find a direct relationship between the molecular weight of chitosan derivatives and the characteristics of the immobilized bromelain obtained by us. Relevant information has been added in Section 3.1. But we certainly agree with the reviewer, that this question is interesting and important for research. This will be the subject of our further study.

Reviewer 2 Report

In this manuscript, the authors address the Bromelain Immobilized on functionalized chitosans and their structural features. They carried out several investigations, providing a reasonable scope for the future application of this method. These prepared Biocatalysts showed significant biological properties, and the obtained results are interesting. Nevertheless, the manuscript requires minor some revisions.

-        In the introduction section, the authors did not provide any detail on the discussion of previous studies carried out with the same type of biocatalyst materials. This should be carefully discussed in this section.

-        More details on the method used for the functionalization of chitosan and the immobilization of bromelain on its structure should be added in the Results and Discussion section.

-        The quality of all figures must be improved.

-        The authors must add a comparison between the prepared material using this method and other reported methods in the literature.

-        However, some typographical and grammatical errors throughout the text must be corrected.

Author Response

We would like to thank the Reviewers for taking their time to read our manuscript. In the view of the constructive criticism by the Reviewers, we have revised the manuscript considerably. All our corrections are highlighted in yellow in the text of the article

In the following, we response to particular concerns raised by the Reviewers in a step-by-step manner.

Reviewer 2, comment 1

In the introduction section, the authors did not provide any detail on the discussion of previous studies carried out with the same type of biocatalyst materials. This should be carefully discussed in this section.

Response

Information related to applications of bromelain jointly with chitosan derivatives were added to the introduction section.

Reviewer 2, comment 2

More details on the method used for the functionalization of chitosan and the immobilization of bromelain on its structure should be added in the Results and Discussion section.

Response

The goal of our work was to investigate the interaction mechanism between bromelain and some functionalized chitosans and evaluate the effect of this interaction on enzyme activity. That’s why we didn’t focus on chitosan modification in the Result and Discussions section. The synthesis conditions, as well as substitution degree of the synthesized products (if this applicable for compound) are present in the Material and Methods section (see 2.2 subsection).   

Reviewer 2, comment 3

The quality of all figures must be improved.

Response

Probably, when adding to the body of the article, the quality of the figures decreased. We provide all the figures in either vector graphics ore in high-resolution format.

Reviewer 2, comment 4

The authors must add a comparison between the prepared material using this method and other reported methods in the literature.

Response

The comparison of the immobilized bromelains obtained with other published data is represent in Fig. 6.

Reviewer 2, comment 5

However, some typographical and grammatical errors throughout the text must be corrected.

Response

Typographical and grammatical errors were corrected